# Picking Up the Threads: Long-Term Outcomes of the Sutured Haemorrhoidopexy: A Retrospective Single-Centre Cohort Study

**DOI:** 10.3390/jcm12010391

**Published:** 2023-01-03

**Authors:** Sara Z. Kuiper, Kayleigh A. M. Van Dam, Merel L. Kimman, Litza Mitalas, Paula G. M. Koot, Jarno Melenhorst, Sander M. J. Van Kuijk, Carmen D. Dirksen, Stephanie O. Breukink

**Affiliations:** 1Department of Surgery, School of Nutrition and Translational Research in Metabolism (NUTRIM), Maastricht University, Universiteitssingel 50, 6229 ER Maastricht, The Netherlands; 2Department of Surgery, Zuyderland Medical Centre, Dr. H. Van der Hoffplein 1, 6162 BG Geleen, The Netherlands; 3Department of Clinical Epidemiology and Medical Technology Assessment, Care and Public Health Research Institute (CAPHRI), Maastricht University Medical Centre, P. Debyelaan 25, 6229 HX Maastricht, The Netherlands; 4Department of Surgery, Maastricht University Medical Centre, Oxfordlaan 10, 6202 AZ Maastricht, The Netherlands; 5Department of Surgery, School for Oncology and Reproduction (GROW), Maastricht University, Universiteitssingel 50, 6229 ER Maastricht, The Netherlands

**Keywords:** haemorrhoidal disease, surgery, sutured haemorrhoidopexy, haemorrhoidopexy

## Abstract

Background: This study aimed to assess the short- and long-term safety and efficacy of the sutured haemorrhoidopexy (SH) in patients with haemorrhoidal disease (HD). Methods: A retrospective study was performed, assessing the following treatment characteristics: number of sutures needed; operation time; perioperative complications; postoperative pain; hospital stay. The short- and long-term postoperative complications, HD recurrence and data on current HD symptoms were assessed according to the Core Outcome Set for HD. Results: Between January 2009 and December 2021, 149 patients with HD underwent a SH. One-hundred and forty-five patients were included, with a mean age of 61 years (±12.8), of which 70 were women (48.3%). Patients were predominantly diagnosed with grade III (37.2%) HD and the median follow-up was nine years (5–11). Perioperative complications occurred in four cases (2.8%). In two patients (1.4%), short-term postoperative complications were reported, and in seven patients (6.2%), long-term complications were reported. The cumulative efficacy in terms of freedom of recurrence was 88.3% (95% CI, 83.1–93.5) at six months, 80.0% (95% CI, 73.5–86.5) at one year, and 67.7% (95% CI, 59.7–75.7) at five years. Conclusions: Sutured haemorrhoidopexy is a safe treatment for patients with HD and can be proposed as a minimally invasive surgical treatment if basic and outpatient procedures fail.

## 1. Introduction

Haemorrhoidal disease (HD) is the most common proctologic disease in the Western world, with prevalence rates of up to almost 40% in the adult population [1]. Basic treatment, including lifestyle, dietary modification and the usage of laxatives, are the first steps in the management of all grades of HD [2]. In patients where basic treatment has not resulted in acceptable symptom reduction, outpatient-procedures such as rubber band ligation (RBL), sclerotherapy and infrared coagulation should be considered as the next treatment step [2,3]. For grade II–IV HD or if outpatient treatment is not sufficient, operative interventions can be proposed. Haemorrhoidectomy is still the gold standard and the most-studied surgical treatment for HD [4]. The rationale of a haemorrhoidectomy is that the prolapsing haemorrhoidal tissue is excised and can no longer protrude from the anal canal. Although the recurrence rates of HD after this procedure are low, around 14% at one-year post-procedure, the main drawback of the haemorrhoidectomy is the painful nature of this operation and the risk of loss of anal function over time [5,6].

Due to these disadvantages, new tissue-sparing techniques were developed, such as stapled haemorrhoidopexy. In comparison to the haemorrhoidectomy, less postoperative pain and a shorter recovery time are reported for this procedure [7]. However, the popularity of stapled haemorrhoidopexy declined over time due to the high costs of the stapler and rare, but possible, severe complications [8,9,10,11].

Another tissue sparing technique is the haemorrhoidal artery ligation (HAL), with or without Doppler-guidance. In this technique, no excision is performed; instead, a pexy suture impedes the arterial inflow to the haemorrhoidal tissue [7]. Studies show that the addition of Doppler does not change the results achieved by the HAL [12,13]. As the HAL mainly focusses on the ligation of the arterial blood flow. A limitation of this technique is that the prolapse is not directly addressed. Since the sensation of tissue prolapse is one of the most frequent symptoms of HD, treatment should be targeted to alleviate this symptom [14]. In line with these tissue-sparing techniques, Pakravan et al. introduced the transanal open haemorrhoidopexy or sutured haemorrhoidopexy (SH), consisting of a suture through the haemorrhoidal complex to lift and fixate the haemorrhoidal complex higher in the anal canal [15]. In addition, a small excision of the anal mucosa is performed to maximise scar tissue. By using only from one to three sutures, the SH is inexpensive in comparison to other surgical techniques. Two studies are known that assess the outcomes of the SH. One prospective study included 38 HD patients and reported that 84% of the patients was pain-free immediately after surgery and no patients required additional surgery after six months [15]. The second study is a German retrospective study assessing SH in 110 patients with a follow-up of over eight years [16]. In 72.7% of the patients, HD symptoms were significantly improved, or even vanished after treatment. The results of the latter study were not reported in the English language, diminishing the impact of the results in scientific society.

The question remains if SH could be a valuable surgical option in the treatment algorithm of HD once outpatient interventions fail. The present study evaluates both short- and long-term outcomes of the SH in patients with symptomatic grade I–IV haemorrhoids by means of a retrospective cohort study.

## 2. Materials and Methods

### 2.1. Patients

All consecutive patients with symptomatic grade I–IV haemorrhoids and operated in our centre between the first of January 2009 and 31 December 2021 were identified for this retrospective study. Patient numbers were retrieved from the electronic patient file (EPF) using the CTcue software package (CTcue b.v.—an IQVIA business, Amsterdam, The Netherlands) and were screened on eligibility. Men and women older than 18 years at the time of their operation, who underwent SH as a treatment for HD, were eligible for inclusion. Eligible patients were asked for written consent by their treating physician before accessing their EPF. Following consent, operative and follow-up data of all patients were retrieved from the EPF.

Additional written informed consent was obtained from patients participating in the telephone interview assessing the current symptoms of HD.

### 2.2. Surgical Procedure

The procedure is displayed in Figure 1. No prophylactic antibiotics or bowel preparation were given. The procedure was performed under general anaesthesia with the patient in the lithotomy position. A proctoscope was used for exposure of the anorectum. The haemorrhoidal tissue was translocated distal in the anal canal with a small surgical clamp. A stitch was placed at the cranial side of the haemorrhoidal complex using 2-0 Vicryl. Distal to this stitch, a one-centimetre strip of mucosa was excised or devitalized with a diathermic device to facilitate scar-tissue formation. Subsequently, the original stitch was completed by performing a second piercing distal to the location of the mucosectomy. After tightening of this suture, the haemorrhoidal tissue was lifted proximally into the anal canal.

This procedure was performed in one to three haemorrhoidal columns depending on the severity of HD. An absorbable haemostatic gelatine tampon was inserted. No postoperative antibiotics were given.

### 2.3. Outcomes and Instruments

#### 2.3.1. Treatment Characteristics and Perioperative Outcomes

Treatment characteristics were collected from the EPF, consisting of the number of sutures needed, operation time (in minutes), perioperative complications, postoperative pain and analgesia use, and hospital stay (in days).

#### 2.3.2. Outcomes According to the Core Outcome Set (COS) for Haemorrhoids

The short- and long-term postoperative complications were assessed according to the European Society of ColoProctology (ESCP) Core Outcome Set (COS) for HD [17].

Short-term postoperative complications consisted of urinary retention and the formation of an abscess, both assessed via physical examination (PE), occurring within or after seven days postoperatively. Long-term postoperative complications were defined as development of anal stenosis, incontinence or anal fistula over a period of one year postoperatively. Both anal stenosis and anal fistula were identified by means of a PE, and incontinence was classified according to the Wexner Incontinence Scale. The Wexner Incontinence Scale contained five questions regarding anal incontinence, describing the type and frequency of the complaint [18]. A higher the score on the scale correlates with an increase in anal incontinence complaints, with a cut-off value of ≥9 or higher indicating a degree of anal incontinence that affects quality of life [19].

All patients were invited for a telephone interview assessing their current symptoms of HD. During this interview, haemorrhoidal symptoms were assessed according to the Patient-Reported Outcome-Haemorrhoidal Impact and Satisfaction Score (PROM-HISS) [20]. First, patients were asked to answer the symptom items of the PROM-HISS. Symptoms included blood loss, prolapse, pain, itching and soiling, and were scored on a Likert scale from 1 (no burden) to 5 (severe burden). Furthermore, the impact of these symptoms on daily activities was probed on a scale from 0 (no impact) to 10 (severe impact), and the satisfaction with treatment was scored on a scale from 0 (not satisfied) to 10 (very satisfied).

#### 2.3.3. Recurrence and Re-Treatment

Next to the COS outcomes, we collected data regarding the recurrence of HD complaints and re-intervention(s) from the EPF. Additional treatment(s) were scored if patients underwent further treatment after SH.

### 2.4. Data Analysis

Patient characteristics were summarized using descriptive statistics. Categorical variables were presented as frequencies with percentages. Continuous variables were presented as mean ± standard deviation (SD) for a normal distribution or as median and first and third quartile for a skewed distribution (IQR). A one-way ANOVA was used to assess the difference between the number of sutures needed and grade of HD. A Kaplan–Meier analysis was performed to analyse efficacy in terms of freedom of recurrence for the entire group, as well as for the subgroups based on HD grade. All statistical analyses were performed in SPSS (IBM SPSS Statistics version 25). Values of *p* < 0.05 were considered statistically significant.

## 3. Results

### 3.1. Patients

In total, 367 consecutive patients were retrieved from the EPF, of which 149 were deemed eligible as they were diagnosed with grade I–IV HD, underwent SH and were older than 18 years at the time of the operation. Four patients did not give consent for access to their EPF. Of the 145 patients, 50 patients gave additional written informed consent for a telephone interview to assess the current symptoms of HD. On average, patients were operated nine years ago (IQR 5–11) and were diagnosed with grade III (37.2%) or IV (29.7%) HD. As only one patient was diagnosed with grade I HD (0.7%), grade I and grade II HD were combined. Patient characteristics of the total cohort (*n* = 145) and the subgroup (*n* = 50) are summarized in Table 1. A study flowchart is shown in Figure 2.

### 3.2. Outcomes

#### 3.2.1. Treatment Characteristics and Perioperative Outcomes

The number of sutures varied between the interventions with a mean of three sutures per procedure. No statistically significant difference was seen in the number of sutures and grade of HD as determined by one-way ANOVA (*p* = 0.991), albeit grade I was excluded (*p* = 0.983). The mean operation time was 18.1 min (±7.7). Perioperative complications occurred in four cases (2.8%); twice including a thrombosed haemorrhoid, once including an anal fissure, and once with the need for a perioperative subsequent haemorrhoidectomy.

Of the patients, 45 (31.0%) out of the 53 patients (36.6%) who reported pain needed analgesia for pain relief. In most cases, the analgesic of choice was oxycodone. Alternatives were combinations of acetaminophen, non-steroidal analgesics and/or oxycodone. The median duration of hospital stay was one day [1–3 days].

Outcomes are shown in Table 2.

#### 3.2.2. Outcomes According to the Core Outcome Set (COS) for Haemorrhoids

Short-term postoperative complications were recorded in two patients (1.4%). Both needed a urinary catheter. No abscesses were documented.

Long-term postoperative complications were seen in seven patients (6.2%), including four patients with faecal incontinence (2.8%), two patients with anal stenosis (1.4%) and one patient who developed a fistula (0.7%). Faecal incontinence was expressed as the Wexner incontinence score, with a mean of 4.75 (±3).

Of the total of 145 patients, a subgroup of 50 patients (34.5%) gave informed consent for a telephone interview. Patients in this subgroup were operated seven years (median, IQR 3–10) before the telephone interview took place. More than half of the patients was still bothered by some feeling of a prolapse from the anus (56.0%), ranging from ‘very little’ to ‘a lot’. Both blood loss and pain were reported in 19 cases (38.0%). About three-quarters of the patients did not experience ‘itching’ or ‘fluid loss’, with ‘itching’ being reported in 13 cases (26.0%) and fluid loss in 12 cases (24.0%). An overview of the responses on the PROM-HISS can be found in Table 3 and Figure 3.

#### 3.2.3. Recurrence and Re-Treatment

Recurrence of complaints after the SH was reported in 49 patients (33.8%), of which 48 patients underwent additional treatment(s). Most of these patients, 37 patients (77.1%), underwent additional RBL treatment(s), with a mean of 2.9 (±2). Fifteen patients underwent a subsequent operative intervention: 12 patients (25.0%) an SH and three patients (6.3%) a haemorrhoidectomy.

The efficacy in terms of cumulative freedom of recurrence was 88.3% (95% CI, 83.1–93.5) at six months, 80.0% (95% CI, 73.5–86.5) at one year and 67.7% (95% CI, 59.7–75.7) at five years, as can be seen in the Kaplan–Meier curve (Figure 4).

The distribution of recurrence according to HD grade can be found in Table 4. The probability of being free from recurrence was 39.9% (95% CI, 15.6–64.2) for grade II, 59.8% (95% CI, 45.9–73.7) for grade III, 62.7% (95% CI, 33.1–92.3) for grade IV after 12 years.

## 4. Discussion

This retrospective cohort study assessing the short- and long-term efficacy and safety of the SH in 145 patients demonstrated a freedom from recurrence of 88.3% (95% CI, 83.1–93.5) at six months, 80.0% (95% CI, 73.5–86.5) at one year and 67.7% (95% CI, 59.7–75.7) at five years. Other studies reported lower recurrence rates for the SH than ours. Pakravan et al. included 38 patients, of whom none required additional surgery after the follow-up of six months [15]. Aigner et al. followed their 40 patients for one year and reported a recurrence rate of 5% in the SH group [12]. Only one randomized controlled trial has been conducted on the efficacy of SH, reporting on 100 patients with a follow-up of two years [21]. In this trial, DG-HAL was compared with suture fixation alone. At two years’ post-procedure, a low recurrence rate of 2.3% in the suture fixation group was seen. Our higher recurrence rate could be explained by our older population. The mean age of our study population was ten years higher than in the above-mentioned studies. It is known that the incidence of HD increases with age and the peak age for HD development is between 45 and 65 years [22,23].

In comparison with the haemorrhoidectomy, the probability of recurrent HD symptoms is higher after an SH. The recurrence rate after a haemorrhoidectomy ranges from around 14 to 16% at one-year post-procedure [5,24]. This lower percentage could be related to the fact that more haemorrhoidal tissue is excised in comparison to the SH, attributing to its more invasive character. As a consequence, it is known that haemorrhoidectomy is associated with a higher complication rate of approximately 10% [25]. In the short term, urinary retention is seen in 2.1–16.4% of patients and local infection in 0.6–1.5% of patients [26,27,28]. Long-term complications can include anal incontinence (15–21.1%), stenosis (2.9–4.7%), fistulas (0.2–1.2%) or other anorectal loss of function [6,26,27,28,29,30,31].

Our study reported only two short-term postoperative complications (1.4%); both concerned a urinary retention, and seven long-term postoperative complications (6.2%), including four cases of incontinence; two cases of anal stenosis, and one case of an anal fistula. Our complication rates are comparable to the study by Gupta et al., who described the mucopexy technique in a large patient cohort (*n* = 616) with a follow-up of one year. In the study of Gupta et al., complications were identified in 9% of the patients, which included the retention of urine, pain needing readmission, bleeding needing readmission, external haemorrhoidal thrombosis, anal tags and pruritus [32].

Besides the higher overall complication rate of haemorrhoidectomy, up to 65% of patients reported moderate to severe pain following this technique, resulting in increased postoperative analgesic use [33,34,35]. In our study, 53 patients (36.6%) reported postoperative anal pain and the majority of these patients required analgesics. Pakravan et al. described the exact same technique and reported that most patients (89%) were free of pain after one-month’s follow-up [15].

Comparing the results from the previously published literature with our data is challenging as the inclusion criteria (i.e., source/target populations) and definitions and timing of the measurement of outcomes vary between different studies, i.e., our study only reported on postoperative pain directly after surgery. The heterogeneity in outcome reporting is one of the downsides of all the previously mentioned studies. For example, the outcome ‘recurrence’ was not defined in all studies, which hampers the possibility of reliably comparing the study results. In this study, we used the definition of recurrence, as stated by the COS for HD [17]. Additionally, other study outcomes also followed the COS for HD, by assessing HD symptoms via the PROM-HISS and recording the predefined short- and long term postoperative complications. Subsequently, the outcomes of this study can be used to compare with future studies on HD treatment and contribute to (inter)national guidelines on HD [2].

This study has some limitations. The retrospective setting of our study should be considered when interpreting the results. As the data were collected from the EPF, which is not originally designed to collect data for research, some information is bound to be missing. Subsequently, selection and recall biases could be possible issues [36]. Moreover, if an outcome is not reported in the EPF, this does not mean that the event did not happen. Hence, the under-reporting of outcomes could have ensued. Additionally, the relatively small sample-size of the subgroup results in estimates that are less precise compared to those of larger groups. Finally, the single centre set-up of this study could also be a limitation, by limiting generalizability.

Despite these limitations, to our knowledge, this is the first study reporting on the short- and long-term efficacy and safety on the SH. It provides real-world evidence that the SH has merit in the treatment algorithm of HD.

As stated in the European Society of ColoProctology guidelines for HD, the haemorrhoidectomy is still the mainstay operation for patients with grade II-III HD and/or in patients who are refractory to outpatient procedures, with the low recurrence rate as its principal selling point [2]. However, in terms of postoperative pain and long-term complications, i.e., incontinence, the SH could serve as a worthy alternative to haemorrhoidectomy. The possible benefits and harms of all treatments should be discussed by doctor and patient as part of the shared decision-making process in the consultation room, allowing for a personalized treatment approach. Future work should include a prospective comparative study assessing the predefined outcomes, according to the COS for HD, to show the actual benefit and comparative value of SH.

## 5. Conclusions

This retrospective study based on real world data study shows that the SH is a safe treatment for HD and can be proposed as a minimally invasive treatment if basic and outpatient procedures are deemed to be ineffective. Its comparative effectiveness to other treatments needs to be evaluated using prospective studies.

## Figures and Tables

**Figure 1 jcm-12-00391-f001:**
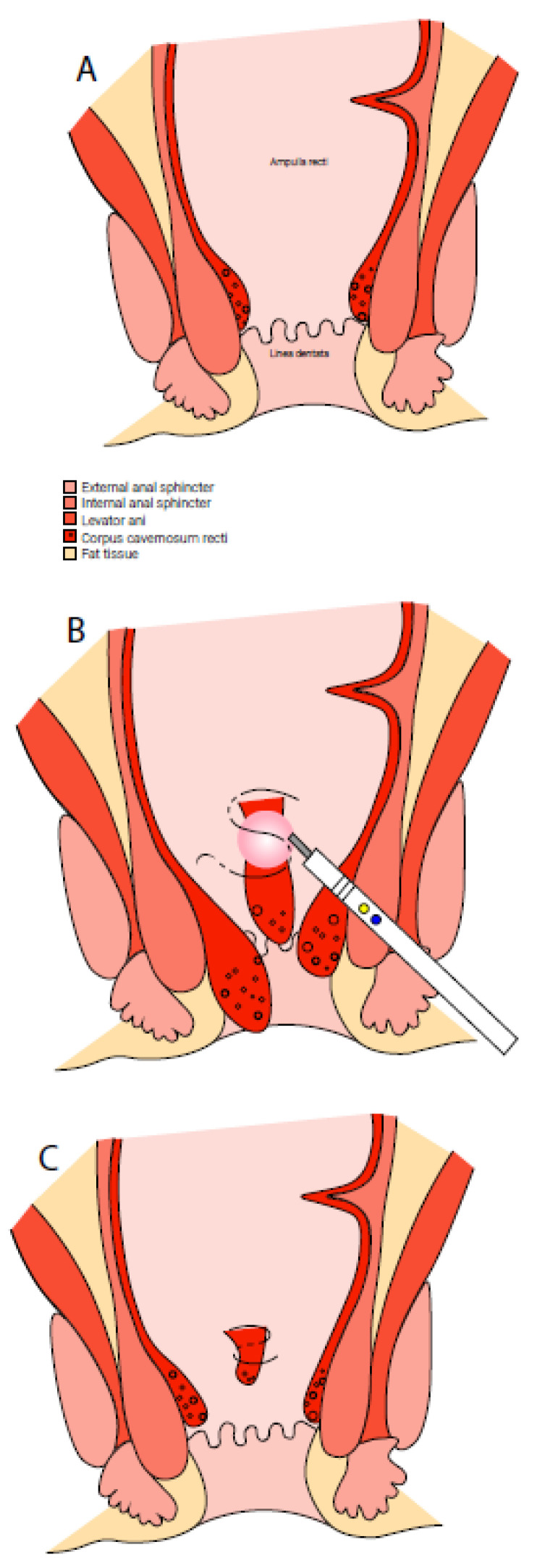
The procedure of the sutured haemorrhoidopexy. (**A**) Healthy situation. (**B**) Haemorrhoidal disease. A stitch is placed at the cranial side of the haemorrhoidal tissue. A mucosectomy is performed with a diathermic device. (**C**) The stitch is tightened, lifting the haemorrhoidal tissue.

**Figure 2 jcm-12-00391-f002:**
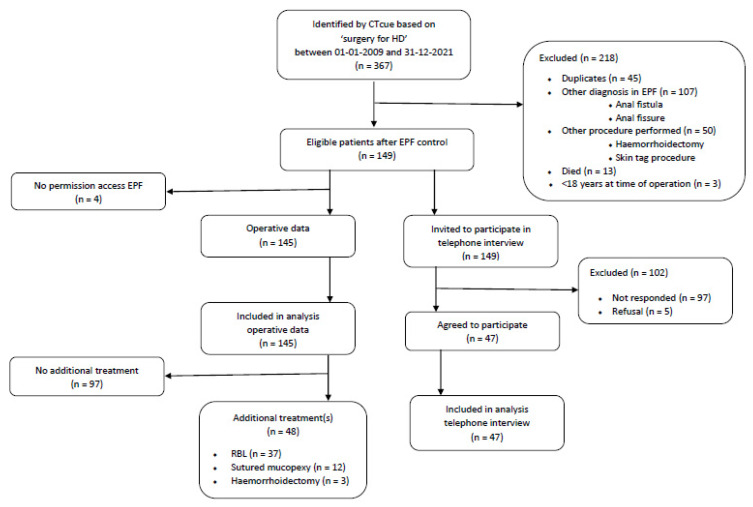
Flowchart of the participating patients.

**Figure 3 jcm-12-00391-f003:**
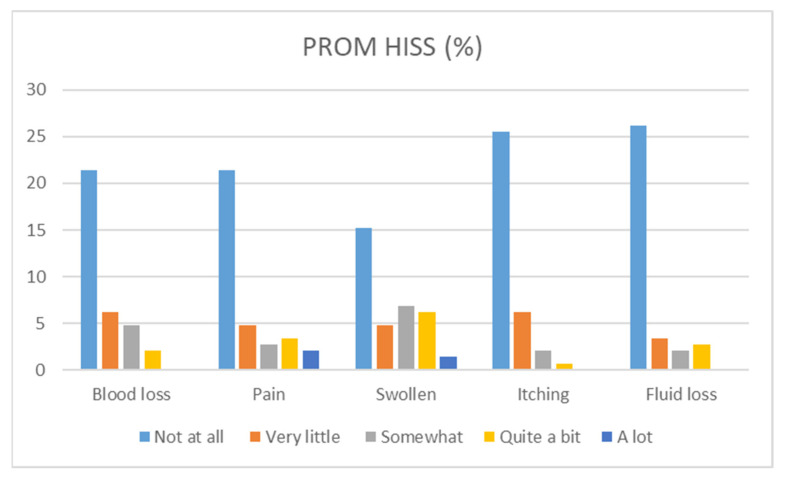
Overview of the responses on the Patient-Reported Outcome Measure-Haemorrhoidal Impact and Satisfaction Score (PROM-HISS).

**Figure 4 jcm-12-00391-f004:**
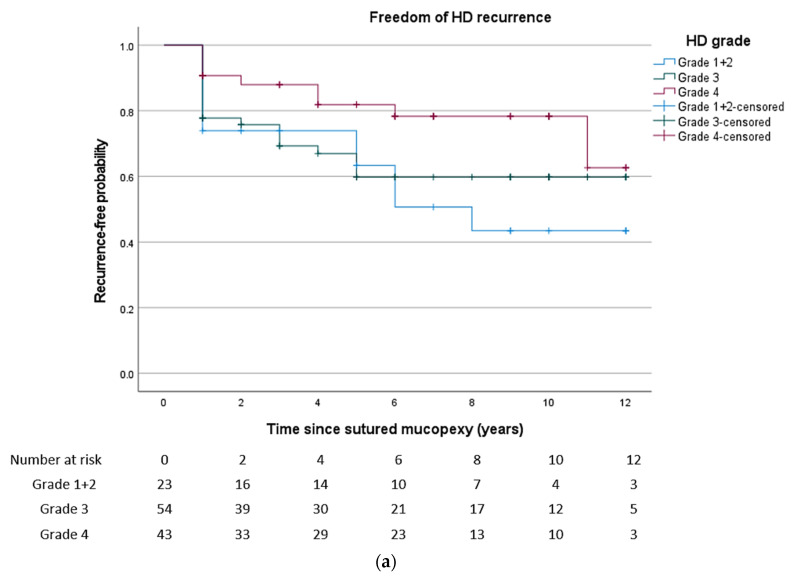
Kaplan–Meier curve of freedom of recurrence. (**a**) According to grade of haemorrhoidal disease. (**b**) According to follow-up time.

**Table 1 jcm-12-00391-t001:** Patient characteristics (*n* = number, y = years).

Characteristics	Total (*n* = 145)	Subgroup ^1^ (*n* = 50)
Female, *n* (%)	70 (48.3%)	21 (42.0%)
Age, mean, ±SD, y	61 ± 12.8	62 ± 11
Years between surgery and follow-up, median (IQR), y	9 [5,6,7,8,9,10,11]	7 [3,4,5,6,7,8,9,10]
*Goligher’s classification*, *n* (%)		
Grade I + II	23 (15.9%)	5 (10.0%)
Grade III	54 (37.2%)	18 (36.0%)
Grade IV	43 (29.7%)	18 (36.0%)
Unknown	25 (17.2%)	9 (18.0%)

^1^ Subgroup with additional data on current symptoms and re-interventions.

**Table 2 jcm-12-00391-t002:** Treatment characteristics and perioperative results (*n* = number, y = years).

Characteristics	Outcome
Number of sutures	3 [1,2,3,4,5,6,7,8]
Operation time, min	18.1 ± 7.7
Perioperative complications	4 (2.8%)
Analgesics needed	45 (31.0%)
*Type of analgesic* ^1^	
Acetaminophen	25 (32.1%)
Non-steroidal analgesics	23 (29.5%)
Oxycodone	27 (34.6%)
Other	3 (3.8%)
Hospital admission, days	1 [1,2,3]

^1^ A combination of analgesics could be given.

**Table 3 jcm-12-00391-t003:** Overview of responses on the PROM-HISS, *n* (%) (PROM-HISS = patient-reported haemorrhoidal-disease-haemorrhoidal impact and satisfaction score, *n* = number).

	Blood Loss	Pain	Swelling	Itching	Fluid Loss
Not at all	31 (62.0%)	31 (62.0%)	22 (44.0%)	37 (74%)	38 (76.0%)
Very little	9 (18.0%)	7 (14.0%)	7 (14.0%)	9 (18.0%)	5 (10.0%)
Somewhat	7 (14.0%)	4 (8.0%)	10 (20.0%)	3 (6.0%)	3 (6.0%)
Quite a bit	3 (6.0%)	5 (10.0%)	9 (18.0%)	1 (2.0%)	4 (8.0%)
A lot	0 (0.0%)	3 (6.0%)	2 (4.0%)	0 (0.0%)	0 (0.0%)

**Table 4 jcm-12-00391-t004:** Distribution of recurrence according to HD grade (HD = haemorrhoidal disease, *n* = number).

Grade HD	(*n* = 145)	Recurrence
Grade I + II	23	11 (47.8%)
Grade III	54	20 (37.0%)
Grade IV	43	9 (20.9%)
Unknown	25	9 (36.0%)

## Data Availability

The data presented in this study are available on request from the corresponding author. The data are not publicly available due to privacy regulations.

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
