# Peer review of "Picking Up the Threads: Long-Term Outcomes of the Sutured Haemorrhoidopexy: A Retrospective Single-Centre Cohort Study"

_jcm, 2023, doi:10.3390/jcm12010391_

Round 1

Reviewer 1 Report

Thank you for the possibility to review the article entitled “Picking up the threads: Long-term outcomes of the sutured 2 haemorrhoidopexy: A retrospective single centre cohort study”.

Overall, the study is well conducted. Limitations are correctly addressed. I would add more in detail the postoperative analgesic treatments. Although this study does not add much to the current literature, the follow-up is rather long.

Author Response

Dear Reviewer,

Thank you very much for taking the time to review our article: “Picking up the threads: Long-term outcomes of the sutured haemorrhoidopexy: A retrospective single centre cohort study”.

  • According to the reviewer’s suggestion regarding the postoperative analgesic treatments, we have added a sentence in the Results section (line 175-176): “In most cases, the analgesic of choice was oxycodone. Alternatives were combinations of acetaminophen, non-steroidal analgesics and/or oxycodone.”

Looking forward to receiving your reply,

happy holidays, Sara Kuiper

Reviewer 2 Report

I read with pleasure and interest the manuscript by SZ Kuiper and colleagues "Picking up the threads: Long-term outcomes of the sutured haemorrhoidopexy: A retrospective single centre cohort study". The paper describes the results obtained in original single-centre retrospective study aimed to assess safety and efficacy of the sutured hemoroidopexy in patients with symptomatic grade I-IV hemorrhoid disease. According to the described, the procedure of interest was associated with low rate of complications and good sustained results. The results are novel and may be interesting for the readers. The paper is well-organized and easy to read. I have only a few minor comments.

It seems that colors chosen to different structures on figure 1 are close to each other. Is it possible to make them more contrast?

Please, ensure that the devices and software used in the study are given together with the name of a manufacturer and a country of origin.

It seems that statistics operating the data of 1 patient could not be correct. May I suggest to combine the data of grade 1 and 2 of hemorrhoidal disease throughout the text/tables?

Author Response

Dear Reviewer,

Thank you very much for taking the time to review our article: “Picking up the threads: Long-term outcomes of the sutured haemorrhoidopexy: A retrospective single centre cohort study”.

  • In the process of constructing Figure 1, we aimed to stay as close to the human anatomy as possible, which is reflected in the choice of colors used. We could change the colors of this Figure, however, this would not be our preference, as this would diminish the Figure’s lifelike appearance.
  • The devices used in the procedure of the sutured haemorrhoidopexy are the proctoscope and 2-0 Vicryl. Both can be of any manufacturer and different hospitals use different manufacturers, based on their individual budget. The charm of the sutured haemorrhoidopexy is that it can be performed with solely these two materials, which makes it a relatively cheap procedure. Regarding the CT cue software, we followed the suggestion of the Reviewer and have added the manufacturer and country of origin in the methods section of the text (lines 85-86): “(…) using the CTcue software package (CTcue b.v. – an IQVIA business, Amsterdam, The Netherlands) (…)”
  • Thank you for your comment. We have adjusted the manuscript based on your suggestion and combined grade I and II haemorrhoidal disease. We have added a sentence in the results section explaining the combination of the groups (line 159-160): “As only one patient was diagnosed with grade I HD (0.7%), grade I and grade II HD were combined.” Furthermore, we combined the two groups (grade I and grade II) in table 1 and 4, and adjusted the Kaplan-Meier curve accordingly.

Looking forward to receiving your reply,

happy holidays, Sara Kuiper